# Pathological Findings in COVID-19 and Non-COVID-19 Vaccine-Associated Lymphadenopathy: A Systematic Review

**DOI:** 10.3390/jcm11216290

**Published:** 2022-10-25

**Authors:** Tzy Harn Chua, Angela Takano

**Affiliations:** 1Department of Anatomical Pathology, Singapore General Hospital, Academia, 20 College Road, Singapore 169856, Singapore; 2Duke-NUS Medical School, 8 College Road, Singapore 169857, Singapore

**Keywords:** COVID-19, vaccine, lymphadenopathy, non-COVID-19, Kikuchi–Fujimoto disease, cancer, metastases

## Abstract

COVID-19 vaccine-associated lymphadenopathy (C19-VAL) is increasingly encountered with the widespread use of the vaccine in controlling the outbreak. We aim to characterize the pathological findings of COVID-19 and non-COVID-19 vaccine-associated lymphadenopathy (NC19-VAL). A search for studies that reported pathological findings in vaccine-associated lymphadenopathy on PubMed and Google Scholar was performed on 11 December 2021. C19-VAL studies were pooled for analysis. These studies were split into clinical lymphadenopathy (CL) and subclinical lymphadenopathy detected on imaging (SLDI) for subgroup analysis. A total of 25 studies were related to COVID-19 vaccines, and 21 studies were included in the pooled analysis. The pooled analysis included 37 patients with a mean age of 47.8 ± 19.1 years old, and 62.2% were females. The mean duration from last vaccination to development of CL/SLDI was 14.5 ± 11.0 days. Most were diagnosed as reactive or negative for malignancy (28/37, 75.5%), followed by Kikuchi–Fujimoto disease (KFD) (3/37, 8.1%), florid lymphoid hyperplasia (2/37, 5.4%), and granulomatous inflammation (2/37, 5.4%). Metastases were reported in two patients with a history of malignancy (2/37, 5.4%). Cases with florid lymphoid hyperplasia and KFD were younger than those with reactive changes. A total of 14 studies were related to non-COVID-19 vaccines. Caseating granulomatous inflammation was reported in BCG vaccine-associated lymphadenopathy, while other vaccines were associated with reactive lymphoid hyperplasia, florid post-vaccinal reactions, and KFD. Although most C19-VAL cases were reported as reactive or negative for malignancy, other diagnoses included florid lymphoid hyperplasia, KFD, and granulomatous inflammation. Metastases were reported in lymphadenopathy of patients with a history of malignancy, who had been incidentally vaccinated. In conclusion, C19-VAL can yield different histopathological diagnoses when sampled, most of which require clinical and radiological correlation for optimal patient management.

## 1. Introduction

The COVID-19 global pandemic, which was caused by a novel strain of coronavirus named severe acute respiratory syndrome coronavirus 2 (SARS-CoV-2), was first reported in December 2019 [1]. At the time of writing, at least 276 million cases, with 5.3 million deaths, have been reported worldwide [2]. In August 2021, the United States Food and Drug Administration approved the use of the Pfizer-BioNTech vaccine, a messenger ribonucleic acid (mRNA) vaccine, in individuals aged 16 years and above [3]. Since then, more than 8 billion doses of vaccine have been administered [2]. The safety and efficacy of the vaccine was evaluated in a clinical trial of 43,548 participants, where it was reported that a two-dose regimen of Pfizer-BioNTech was 95% effective in preventing COVID-19 [4], although a decline in vaccine efficacy to 91.3% after 6 months of follow-up has been reported [5]. In the landmark trial by Polack et al. [4], 0.3% of vaccine recipients reported lymphadenopathy amongst other adverse events, including reduced appetite, lethargy, malaise, and night sweats. Higher rates of axillary swelling or tenderness were also reported in the Moderna vaccine [6], where 11.6% and 16.0% experienced these symptoms after the first and second dose, respectively.

Post-vaccinal lymphadenopathy due to reactive changes in the lymph nodes is well-described [7], and it was previously reported in a number of vaccines, including bacillus Calmette–Guerin (BCG) [8], hepatitis B [9], human papillomavirus [10], and tetanus [11], amongst several others. However, SARS-CoV-2 vaccines are the first mRNA vaccines to be approved for clinical application. SARS-CoV-2 vaccines work primarily through the delivery of an mRNA into cells, where the mRNA is translated into a target protein, against which the immune system will mount an immune response, which includes antigen presentation in the regional lymph nodes, the priming of CD4+ and CD8+ T cells, germinal center reaction, and finally the formation of affinity-matured memory B cells and antibody-secreting long-lived plasma cells [12]. In a study of antigen-specific B cells in the peripheral blood and fine-needle aspirates (FNA) of draining lymph nodes from 14 participants that had received two doses of BNT162b2 (Pfizer-BioNTech) vaccine [13], a strong plasmablast response in the blood and a robust germinal center B cell and plasmablast response in the aspirates were reported, and they were persistent for at least 12 weeks after the second dose of vaccine, which is an indicator of developing a potent humoral response [14].

COVID-19 vaccine-associated lymphadenopathy (C19-VAL) is often detected either clinically as a palpable and/or painful lump or on screening and follow-up radiological studies, including breast ultrasound [15] and positron emission tomography/computed tomography (PET/CT) [16], where hypermetabolic axillary lymph nodes occurred in 37.0% of patients who had received the COVID-19 vaccine. A meta-analysis [17] of 68 cases of C19-VAL with imaging findings reported that the median duration from vaccination to development of lymphadenopathy was 12 and 5 days for the first and second dose, respectively, of which 42.6% showed diffuse or focal cortical thickening on ultrasound. Furthermore, an increased uptake on PET/CT was reported in 26.4%, with a mean maximum dimension of lymph nodes reported to be 20.9 mm [17]. Although some of these radiological findings may favor a reactive lymphadenopathy, studies have also reported suspicious ultrasound findings in lymphadenopathy after COVID-19 vaccination in patients being followed up for skin cancer [18], raising a diagnostic conundrum between a reactive and a malignant process. Similar diagnostic dilemmas occur in breast cancer screening, which involves a significant number of healthy women [15].

Histopathological and cytopathological findings, obtained through procedures including FNA and core needle biopsy (CNB), are only reported in a proportion of C19-VAL, and these findings remain to be characterized. Hence, in this review, we aim to identify the current studies that have reported pathological findings of C19-VAL, and NC19-VAL for comparison.

## 2. Material and Methods

### 2.1. Search Strategy

We performed a literature search in accordance with the PRISMA statement [19], through PubMed and Google Scholar, on 11 December 2021. The keywords included ‘vaccine’ and ‘lymphadenopathy’. We placed no restrictions on the year of publication. We searched reference lists of full-text articles for snowballing of additional studies that were not identified in the initial search.

### 2.2. Inclusion and Exclusion Criteria

We included studies that reported histopathological and/or cytological findings in vaccine-related lymphadenopathy, without any restrictions on the type of studies. We also included preprint articles if they met the abovementioned criteria. We excluded review, recommendation, and non-English articles.

### 2.3. Data Extraction and Analysis

We included studies that reported the clinical history, investigation, and pathological findings of individual patients in the pooled analysis. We extracted the pertinent details from each study, and these included: the type of publication, number of patients with pathological findings, age and sex of patients, significant history, type and dose of most recent vaccine, whether the lymphadenopathy was detected clinically or on imaging, site of vaccine, other clinical symptoms, duration from last vaccination to lymphadenopathy, laterality of lymphadenopathy compared with site of vaccination, site of lymphadenopathy, largest dimension of lymph node, ultrasound findings, additional radiological findings, indication for aspiration or biopsy, type of procedure performed, pathological diagnosis, and management and outcome. We performed pooled analysis of the variables and expressed these variables either in means and standard deviations or in percentages.

As patients presenting with clinical lymphadenopathy (CL) represent a distinct population of patients compared with those with subclinical lymphadenopathy detected on imaging (SLDI), we analyzed these two groups separately. SLDI patients were detected on routine imaging follow-up for conditions including a history of malignancy, or screening for malignancy such as breast cancer. We also split patients with CL into subgroups according to their pathological diagnosis (reactive changes or negative for malignancy, florid lymphoid hyperplasia, KFD, etc.), and analyzed separately. We used the *t*-test to compare the means and standard deviations (www.medcalc.org, accessed on 4 January 2022). We used the chi-square to compare the percentages. We considered a *p*-value of less than or equal to 0.05 as statistically significant.

## 3. Results

### 3.1. Literature Search

An initial search of PubMed and Google Scholar yielded a total of 618 results, with 77 duplicates, and 541 results were screened (Figure 1). After excluding 428 articles, 108 full-text articles were obtained and assessed for eligibility. After excluding 74 articles for the reasons stated in Figure 1, and with the addition of 5 articles identified from the reference lists of articles, 39 articles were included in this review and 21 COVID-19 studies were included in the pooled analysis. A total of 25 studies [18,20,21,22,23,24,25,26,27,28,29,30,31,32,33,34,35,36,37,38,39,40,41,42,43] were relevant to the COVID-19 vaccine and 14 studies [7,8,9,10,44,45,46,47,48,49,50,51,52,53] were related to non-COVID-19 vaccines.

### 3.2. COVID-19 Vaccine-Associated Lymphadenopathy Study Characteristics

The 25 studies were published by authors from countries including Germany [18] (*n* = 1), Israel [22,37] (*n* = 2), Japan [26] (*n* = 1), Portugal [20] (*n* = 1), Qatar [31] (*n* = 1), Singapore [32,33] (*n* = 2), South Korea [27,30,39] (*n* = 3), Spain [23,24,41] (*n* = 3), Switzerland [21] (*n* = 1), the United Kingdom [36] (UK) (*n* = 1), and the United States of America [25,28,29,34,35,38,40,43] (USA) (*n* = 8). Most studies are case reports (*n* = 12, 48.0%), followed by case series (*n* = 9, 36.0%), retrospective studies (*n* = 3, 12.0%), and not reported in one study (4.0%).

### 3.3. Pooled Analysis of COVID-19 Vaccine-Associated Lymphadenopathy

Findings from the pooled analysis are summarized in Table 1. Pooled analysis of 21 studies [18,20,21,25,26,27,28,29,30,31,32,33,34,35,36,37,38,39,40,41,42] included 37 patients with a mean age of 47.8 ± 19.1 years old, of which 23 were female (23/37, 62.2%). The largest proportion of patients (9/37, 24.3%) had no prior medical history, followed by 21.6% (8/37) with a history of melanoma, and 18.9% (7/37) with a history of breast cancer. The other patients had a history of lung cancer, appendix neuroendocrine tumor, Merkel cell carcinoma, and renal cell carcinoma. The largest proportion of C19-VAL occurred after the first dose (13/37, 35.1%), followed by the second dose of Pfizer-Bio-NTech vaccine (7/37, 18.9%). Other vaccines that were reported in these patients included Moderna, AstraZeneca, Vaxzevria, and CureVac.

The mean duration from the last vaccination to the development of CL or SLDI was 14.5 ± 11.0 days. Most cases of lymphadenopathy (23/37, 62.2%) occurred ipsilateral to the site of vaccination, while two cases (2/37, 5.4%) were contralateral, with the laterality not reported in the remaining cases. With the exclusion of Hagen et al. [21] due to the lack of individual sites of lymphadenopathy reported for each patient, most cases (18/32, 56.3%) had axillary lymphadenopathy, followed by 21.9% (7/32) with supraclavicular lymphadenopathy, and 12.5% (4/32) with cervical lymphadenopathy. Additional clinical symptoms aside from lymphadenopathy were reported in these cases. Overall, six patients (6/37, 16.2%) experienced fever [25,30,33,34], while four patients (4/37, 10.8%) also reported pain [26,27,28,29]. The other described symptoms included fatigue or malaise (2/37, 5.4%), myalgia (2/37, 5.4%), dysphagia (1/37, 2.7%), chills (1/37, 2.7%), and other symptoms such as vomiting (2/37, 5.4%).

The mean largest dimension of lymph node reported clinically or on radiologic studies was 20.8 ± 13.3 mm. Abnormal lymph node findings were reported using ultrasound in 29.7%, computed tomography/magnetic resonance imaging (CT/MRI) in 29.7%, and PET/CT in 21.6%. In terms of indication for aspiration or biopsy, 40.5% (15/37) of these procedures were performed for suspicion of malignancy, followed by other indications including suspicion of lymphadenitis and/or KFD (2/37, 5.4%), palpable mass (2/37, 5.4%), and patient’s preference (2/37, 5.4%). Other indications were family history of malignancy (1/37, 2.7%) and further oncologic management (1/37, 2.7%). Most reported cases had either a CNB (12/37, 32.4%) or excisional biopsy (12/37, 32.4%) performed for the pathological examination of lymphadenopathy, followed by FNA (9/37, 24.3%). The other procedures that were performed included surgical resection and complete lymphadenectomy [18].

Most cases of lymphadenopathy that were sent for pathological examination were diagnosed as reactive or negative for malignancy (28/37, 75.5%). Other histopathological findings were as follows: florid lymphoid hyperplasia in two patients (2/37, 5.4%); KFD in three patients [31,33] (3/37, 8.1%); granulomatous inflammation in two patients [18,42] (2/37, 5.4%); and metastases in two patients [18] (2/37, 5.4%). Further details are elaborated in the following sections.

### 3.4. Clinical Lymphadenopathy (CL)

A total of 15 studies reported pathological findings of C19-VAL. There was a total of eight case reports [20,25,26,27,30,31,32,34], five case series [21,24,28,29,33], one retrospective study [22], and one study type was not reported. Our findings are summarized in Table 2.

Three studies [22,23,24] were not included in the pooled analysis as individual patient data was not reported. Each one of these three studies reported five to eleven patients, of which those in Faermann et al. [22] had a history of breast cancer or were BRCA carriers, and included patients with both CL and SLDI. The vaccines that were used in these studies included Pfizer-Bio-NTech and Moderna. The lymphadenopathy was ipsilateral to the site of vaccination and occurred in the axillary and supraclavicular regions. Further investigation of the lymphadenopathy was performed due to a suspicion of malignancy [22,23]. In terms of pathological findings obtained by ultrasound-guided core biopsy or FNA, these three studies reported reactive findings, with Felices-Farias et al. [23] describing reactive paracortical and interfollicular hyperplasia and Fernandez-Prada et al. [24] reporting reactive lymphocytic infiltrate and active germinal centers.

The pooled analysis of 18 cases with CL showed a mean age of 37.8 ± 15.6 years old, with 50.0% (9/18) females (Table 1). Half of these patients had no prior medical history, while three patients (3/18, 16.7%) had prior non-neoplastic medical history, including asthma, eczema, and hypothyroidism in Tintle et al. [34], steroid-dependent minimal-change renal disease in Soub et al. [31], and diabetes mellitus and hypertension in Tan et al. [33]. The other cases had a family history of breast cancer [20,29], lung cancer [21], and appendix neuroendocrine tumor (NET) [21].

Most cases of lymphadenopathy occurred ipsilaterally to the site of vaccination, with contra-laterality reported in two cases (2/18, 11.1%) [21,30]. The most common site of lymphadenopathy was the supraclavicular region (6/13, 46.2%), followed by the axillary (4/13, 30.8%) and cervical regions (3/13, 23.1%). The most reported associated symptoms included fever (6/13, 46.2%), pain (4/13, 30.8%), as well as other symptoms mentioned in Table 1 and Table 2. The mean largest dimension of lymph node reported was 21.1 ± 14.7 mm.

In terms of radiological findings, eight patients (8/18, 44.4%) had abnormal CT findings, which included ‘irregular thickening and inflammation in the sternocleidomastoid area’ [25] and conglomerated lymph nodes with perinodal infiltration [30], while other CT findings mainly reported that the lymph nodes were enlarged. Seven patients (7/18, 38.9%) had abnormal US findings, which included the loss of a defined hilum or a partially detectable hilum [20,21,26,27,32], and ill-defined borders [27]. Two patients (2/18, 11.1%) had abnormal PET/CT findings with increased FDG uptake [21].

FNA and CNB were most frequently performed for further pathological investigation, with the most commonly reported indication being suspicion of malignancy (8/18, 44.4%). Most cases of CL were reported as either reactive or negative for malignancy (13/18, 72.2%), two cases (2/18, 11.1%) were reported to have florid lymphoid hyperplasia, and KFD was reported in three patients [31,33] (3/18, 16.7%). Notably, Cardoso et al. [20] reported atypical lymphoid findings using FNA; however, the subsequent biopsy disclosed reactive follicular hyperplasia. Similarly, Larkin et al. [28] reported that there was a focal increase in Epstein–Barr Virus (EBV)-positive cells, with other findings suggestive of a prior infection. In the two cases of florid lymphoid hyperplasia [28,34], Larkin et al. [28] also reported that there was progressive transformation of germinal centers, while Tintle et al. [34] reported Langerhans cell hyperplasia. The management and outcomes of CL are outlined in Table 2.

Cases diagnosed with reactive changes or negative for malignancy were compared with those diagnosed with florid lymphoid hyperplasia and KFD. Cases of florid lymphoid hyperplasia and KFD had a mean age of 18.0 ± 5.0 and 23.3 ± 7.5 years old, respectively, which are significantly younger than those diagnosed with reactive changes or as negative for malignancy (44.2 ± 13.1 years old, *p* = 0.02). The duration from the last vaccination to CL in the KFD cases was 20.7 ± 10.5 days, which is significantly longer than those diagnosed with reactive changes or as negative for malignancy (10.9 ± 6.3 days, *p* = 0.048). There was no statistically significant difference in the duration from last vaccination to CL between those diagnosed with reactive changes or as negative for malignancy (10.9 ± 6.3 days), and those with florid lymphoid hyperplasia (10.5 ± 3.5 days). The largest dimension of lymph node did not differ significantly amongst these three diagnoses (reactive changes or negative for malignancy: 22.1 ± 18.2 mm, florid lymphoid hyperplasia: 15.5 ± 5.5 mm, KFD: 22.3 ± 7.1 mm).

### 3.5. Subclinical Lymphadenopathy Detected on Imaging (SLDI)

The findings are summarized in Table 3. A total of eleven studies reported the pathological findings of SLDI, and these included five case series [18,29,36,38,39], four case reports [35,40,41,42], and two retrospective studies [37,43]. Robinson et al. [43] reported a breast cancer patient with SLDI and a biopsy that was negative for malignancy; however, this report was excluded from the pooled analysis as individual patient data was not reported. There were 19 patients identified from these studies with a mean age of 57.2 ± 17.3 years old, which is significantly older than the cases with CL (37.8 ± 15.6 years, *p* = 0.001). Most of these patients were females (14/19, 73.7%). All patients had a prior history of malignancy, with a history of melanoma [18,40,42] in 42.1% (8/19), breast cancer [29,36,37,38,39] in 36.8% (7/19), and Merkel cell carcinoma [18] in 10.5% (2/19), with other malignancies including cecum-appendix NET [41] and renal cell carcinoma [35].

Most cases were associated with the Pfizer-Bio-Ntech vaccine, with equal proportions associated with the first and second dose (31.6%). The other vaccines implicated in these patients with SLDI are presented in Table 1 and Table 3. The mean duration from last vaccination to SLDI was 16.5 ± 12.9 days; however, this was not significantly different from the mean duration from last vaccination to CL (12.5 ± 7.9 days). Most cases of SLDI occurred ipsilateral to the site of vaccination (13/19, 68.4%), while the site of the remaining cases was not reported. The axillary region was the most commonly reported site of SLDI (14/19, 73.7%), and this is significantly higher than that reported in CL (4/13, 30.8%, *p* = 0.018). A total of 5.3% (1/19) of SLDI occurred in the supraclavicular region, which is significantly lower than that reported in CL (6/13, 46.2%, *p* = 0.0069). Of note, Trikannad et al. reported SLDI in the mediastinum [42].

The largest dimension of the lymph node was 19.7 ± 2.9 mm, which is not significantly different from that reported in CL. In terms of radiological findings, 31.6% (6/19) had abnormal PET/CT findings, while 21.1% (4/19) had abnormal US findings and 15.8% had abnormal CT/MRI findings. Abnormal PET/CT findings included FDG uptake or hypermetabolic lymph nodes [29,37,40,41,42], while abnormal US findings included thickened cortex [36,39]. Abnormal CT/MRI findings included a length/width ratio of less than 1.5 [39], cortical thickening [39], and asymmetricity [38].

The indications for aspiration or biopsy are summarized in Table 1 and Table 3. Excision biopsy was performed in 36.8% (7/19), CNB in 26.3% (5/19), and FNA in 10.5% (2/19). The other procedures (4/19, 21.1%) included complete lymphadenectomy and surgical resection.

The majority of SLDI was reactive and/or negative for malignancy (15/19, 78.9%). However, granulomatous inflammation was reported in two cases [18,42] (2/19, 10.5%) and metastases were reported in two cases (2/19, 10.5%) [18]. Trikannad et al. [42] reported non-caseating granulomas in the mediastinal FNA of a 57-year-old female with a history of melanoma. Placke et al. [18] reported a sarcoid-like reaction in a patient with a history of melanoma who underwent complete lymphadenectomy. Unfortunately, this patient experienced post-operative lymphorrhea requiring multiple sclerotherapies [18]. Placke et al. [18] described two patients (without further elaboration of whether these two patients had melanoma or Merkel cell carcinoma) in whom ultrasound of the SLDI could not exclude malignancy. Histopathological examination confirmed metastatic disease in each case.

### 3.6. Non-COVID-19 Vaccine-Associated Lymphadenopathy

Findings are summarized in Table 4. A total of 14 studies that reported pathological findings in NC19-VAL were identified, and these included nine case reports [9,10,11,44,45,47,48,50,51], three retrospective studies [7,8,53], one prospective study [46], and one case series [49]. Seven studies [8,9,44,45,46,47,53] reported on the BCG vaccine, while the others included hepatitis B [9], H1N1 [48], HPV [10,49], Japanese encephalitis virus (JEV) [10], measles [50], rubella [51], and tetanus vaccines [11]. Hartsock et al. [7] reported a range of vaccines including smallpox, cholera, typhus, tetanus, pertussis, Salk (polio), and influenza.

The BCG studies predominantly described caseating granulomatous inflammation either on biopsy [8,9,45,47] or aspiration [8,46,53] of the involved lymph nodes. Of note, Dotlic et al. [9] reported a 2-week-old male with inguinal lymphadenopathy who received both the BCG and hepatitis B virus vaccines, with an initial FNA showing atypical lymphoid cells that were suspicious of lymphoma. Subsequently, the excision biopsy showed an effacement of nodal architecture, with an atypical T cell proliferation that showed an active cytotoxic phenotype, as well as a high proliferative index of 90% [9]. This case was initially diagnosed as a T cell lymphoma, possibly with a lymphoblastic subtype; however, further immunohistochemistry was negative for immature T cell markers, including TdT, CD34, and CD117 [9]. This case was subsequently diagnosed as BCG lymphadenitis with a reactive hyperimmune post-vaccinal reaction [9].

Lymphadenopathy associated with other vaccines showed mainly reactive lymphoid hyperplasia [7,49,50]; however, florid reactions to these vaccines may raise a concern for lymphomas. In the case of H1N1 vaccine-associated lymphadenopathy reported by Toy et al. [48], there were CD30-positive immunoblasts as well as large cells that showed a resemblance to Hodgkin cells, raising the differential diagnosis of Hodgkin lymphoma. This case was subsequently diagnosed as post-vaccinal lymphadenitis. White et al. [11] reported a case of tetanus-associated lymphadenopathy in a 50-year-old female, with excisional biopsy showing sheets of small-to-medium-sized lymphocytes and a flow cytometry study interpreted as atypical T cell population; however, the diagnosis was subsequently reviewed to ‘pseudolymphomatous florid proliferation of CD4 + T cells in response to tetanus toxoid immune stimulation’. KFD was reported by Watanabe et al. [10] in a 14-year-old female who received HPV and JEV vaccines.

## 4. Discussion

In this systematic review, we identified 25 studies that reported the pathological findings of C19-VAL, with 21 studies subsequently included in the pooled analysis, and 14 studies that reported pathological findings of NC19-VAL for comparison. The pooled analysis of the 21 C19-VAL studies included 37 patients with a mean age of 47.8 ± 19.1 years old, of which 62.2% were females. The mean duration from the last vaccination to the development of CL or SLDI was 14.5 ± 11.0 days. Most cases were diagnosed as reactive or negative for malignancy (75.5%), followed by KFD (8.1%), florid lymphoid hyperplasia (5.4%), and granulomatous inflammation (5.4%). Metastases were reported in two patients (5.4%) who had a history of malignancy.

Furthermore, cases with florid lymphoid hyperplasia and KFD were significantly younger than those with reactive changes or negative for malignancy. The duration from last vaccination to CL in KFD cases was significantly longer than those with reactive changes or negative for malignancy. The axillary region was the most commonly reported site of biopsy or FNA for SLDI while the supraclavicular region was the most commonly reported site of biopsy or FNA for CL. For NC19-VAL, caseating granulomatous inflammation was reported in BCG vaccine-associated lymphadenopathy, while other vaccines were associated with reactive lymphoid hyperplasia, florid post-vaccinal reactions, and KFD.

To the best of our knowledge, this is the first systematic review to characterize the pathological findings in C19-VAL. Two key patient populations are implicated: patients with CL and patients with SLDI, who have a history of malignancy and are on active follow-up. Most of the patients with CL had other symptoms and abnormal imaging findings, which may raise a suspicion of conditions other than post-vaccinal lymphadenitis. Regarding histopathological findings, patients with florid lymphoid hyperplasia and KFD were significantly younger than those with reactive changes or negative for malignancy, and patients with KFD developed lymphadenopathy significantly later than those with reactive changes or negative for malignancy. Although patients with KFD have been reported to be younger [54], these findings were based on a limited sample size and need to be interpreted in the appropriate context. Among patients with SLDI, although most cases were histologically diagnosed as reactive or negative for malignancy, granulomatous inflammation and metastases were also reported. Of note, metastases were reported in 2 out of 19 patients (10.5%) in this population. Given a clinical history of a previous malignancy, this differential diagnosis needs to be considered in any patient with C19-VAL. Despite the association of the COVID-19 vaccine with lymphadenopathies diagnosed histologically as KFD, and granulomatous inflammation, it remains unclear whether these conditions are related to the vaccine. The etiology and pathogenesis of KFD remains unclear, with viruses being postulated to be a key inciting agent [55,56]. On the other hand, the cases that were found to have metastatic lymphadenopathy were previously known to have a primary malignancy.

Several systematic reviews investigating the imaging findings in C19-VAL have been performed. Garreffa et al. [57] reported the incidence of clinical and subclinical lymphadenopathy to range from 14.5% to 53% of 2057 patients, and the lymphadenopathy persisted beyond 6 weeks in more than a quarter of these patients. Treglia et al. [16] performed a meta-analysis of 2354 patients who underwent PET/CT after COVID-19 vaccination and reported a prevalence of hypermetabolic axillary lymph nodes in 37% of these patients. Keshavarz et al. [17] performed a pooled analysis of 68 cases of C19-VAL and reported that cortical thickening was seen on the US in 42.6%, and other findings included preserved nodal hilar fat and necrotic patterns. The mean maximum dimension of the lymph nodes reported in the imaging modalities was 20.9 ± 5.8 mm, which is similar to what we have found in this systematic review. A systematic review reported that the cases of C19-VAL detected in patients undergoing follow-up PET/CT were ipsilateral to the vaccine injection site [58]. Brown et al. [59] also provided a narrative review of the imaging findings and reported that loss of normal fatty hilum can be expected on ultrasound. Numerous guidelines and recommendations [15,57,59,60] have been proposed for the management of C19-VAL. A case-by-case patient-centered approach has been suggested in determining whether further investigations or management are warranted in patients with a history of malignancy [15,57]. Diagnostic algorithms have also been suggested for better assessment of axillary lymphadenopathy [61,62], and follow-up with PET/CT [63].

In light of these current findings, it appears that the pathological findings of C19-VAL and NC19-VAL are similar. Traditionally, vaccines are classified as live vaccines, non-live vaccines, as well as viral vectors, RNA, DNA, and virus-like particles vaccines [64]. With the exception of the BCG vaccine that induces T cell responses (cellular immunity), all other routine vaccines confer immune protection through the production of antibodies mediated by B cells (humoral immunity) [64]. Once the vaccine antigen is introduced to the immune system, it is transported to the draining lymph nodes where antigen presentation activates T cells, which subsequently activates B cells, leading to a cascade of events that ultimately result in production of short-lived plasma cells that secrete antibodies in the first 2 weeks after vaccination, and memory B cells and long-lived plasma cells that produce antibodies for decades [64].

In this study, most of the cases were associated with the use of the Pfizer-Bio-Ntech vaccine, with more than one-third associated with the first dose. Numerous COVID-19 vaccines have been developed, and these include DNA and RNA vaccines (Pfizer-Bio-NTech, Moderna), adenoviral-vectored vaccines (AstraZeneca) and whole-cell-inactivated vaccines (Sinovac, Sinopharm) [65]. These vaccines are designed to induce an immune response that is mediated by neutralizing antibodies against the SARS-CoV-2 spike protein [65]. The immune responses induced by COVID-19 vaccines are similar to non-COVID-19 vaccines [12,65,66]. In addition to neutralizing bodies, T cell responses are also implicated in conferring protection, although the exact mechanisms remain undetermined and may influence whether booster doses are necessary or not in the future [65]. The BCG vaccine has also been proposed to be involved in the induction of COVID-19 vaccine immune responses through ‘trained immunity’, where monocytes and natural killer cells undergo epigenetic changes to mount an enhanced response against pathogens [67,68].

COVID-19 vaccines have also been associated with lymphoproliferative disorders [69,70,71] and hyperinflammatory syndromes [72].

Goldman et al. [71] described a case of a 66-year-old man who presented with moderate asthenia and mild inflammatory syndrome without abnormal blood cell counts 6 months after the second dose of an mRNA COVID-19 vaccine. Using PET/CT, he was found to have hypermetabolic adenopathies above and below the diaphragm. Biopsy demonstrated angioimmunoblastic T cell lymphoma (AITL) and next-generation sequencing (NGS) of the biopsy showed mutations, such as DNMT3A and TET2, which correlate with clonal hematopoiesis [73]. Interestingly, he was later administered a booster dose of the same vaccine in the right deltoid area in order to prepare him to receive chemotherapy. A few days later, he developed swelling of the right cervical lymph nodes and a second PET/CT demonstrated increased avidity in all lymphadenopathies and the appearance of new lesions. He was treated immediately with combination chemotherapy and anti-CD30 monoclonal antibody, with a reduction in disease within 2 weeks. Since the presence or absence of initial disease before the vaccination could not be ascertained in this case, the authors highlighted the rapid progression of the recently diagnosed AITL after the booster vaccine rather than implicating the vaccine in the development of the disease itself. They invoked a possibility of interaction of already malignant T cells with the vaccine mechanisms of stimulation of T helper cell immunity.

Sekizawa et al. [69] reported the case of an 80-year-old Japanese female who developed a right temporal mass after the administration of the first COVID-19 vaccine, which was also associated with multiple lymphadenopathies in other sites, including the cervical and supraclavicular regions. This was subsequently diagnosed as a marginal zone lymphoma [69], reinforcing the importance of considering a neoplastic etiology in the differential diagnosis of lymphadenopathies associated with COVID-19 vaccines. Tang et al. [70] reported a case of a 51-year-old male heart transplant recipient who presented with a mediastinal mass one week after receiving the first dose of a COVID-19 vaccine, which was diagnosed as an Epstein–Barr virus (EBV)-positive diffuse large B-cell lymphoma. The authors hypothesized that the COVID-19 vaccine may reactivate latent EBV infection, thus contributing to the development of a neoplastic process [70]. mRNA COVID-19 vaccines have also been associated with immune dysregulation resulting in hemophagocytic lymphohistiocytosis, mainly in patients with pre-existent autoimmune disorders [72]. It is important to recognize these clinical associations with mRNA COVID-19 vaccines in order to be able to perform a risk/benefit analysis in each specific circumstance. Similarly, further research is needed to establish the mechanisms of interaction of the immune response to mRNA COVID19 vaccination and pre-existent immune alterations as well as possible neoplastic predispositions, such as mutations that may correlate with clonal hematopoiesis.^71^ Further research is needed to elucidate the possible mechanisms of pathogenesis and tumorigenesis in these contexts.

In November 2021, the SARS-CoV-2 variant B.1.1.529, also named Omicron, was designated by the World Health Organization as a variant of concern, with an increased risk of reinfection with Omicron in those who had prior COVID-19 [74]. The BNT162b2 (Pfizer-Bio-NTech) vaccine was evaluated in the setting of hospitalized patients with the Omicron variant of COVID-19 [75]. The vaccine effectiveness was reported to be 70% and the authors [75] suggested that a third booster dose of vaccine may be warranted to improve effectiveness, as the vaccine effectiveness was reported to be 93% for hospital admission, 92% for severe disease, and 81% for COVID-19-related mortality [76].

This study is limited by its small sample size. Larger sample sizes may allow the characterization of patients who developed a florid lymphoid hyperplasia or KFD. Furthermore, none of the patients in this review had a history of, or a clinical suspicion for, a lymphoproliferative disorder. As such, the interpretation of these results in a patient that recently received a COVID-19 vaccine and that is suspicious for a lymphoproliferative disorder needs to be performed cautiously, although there have been reported associations between COVID-19 vaccines and lymphoproliferative disorders as mentioned above. Additionally, the majority of the included studies did not detail the pathological findings aside from a diagnostic line, and this further limits a deeper analysis of these results. It would also be of interest to determine whether each vaccine is associated with a different rate of diagnoses with larger sample sizes. There is also an inherent selection bias, especially for SLDI cases, as these cases of incidental lymphadenopathy would likely be more frequently detected by surveillance imaging. However, considering that only a proportion of C19-VAL is further investigated with FNA or biopsy, this review has the largest number of C19-VAL with pathological findings. As a result of the methodology of pooling individual patient data from published studies, there is heterogeneity in the patient population, demographics, and clinical characteristics. Additionally, most of the studies identified did not have a detailed report of the pathological findings, including for the further ancillary immunohistochemical studies that were performed.

This study does not include specifically pediatric patients, since the vaccine was only later approved to be used in children aged 5 and above and recently for 6 months and above.

## 5. Conclusions

C19-VAL is gaining recognition with the widespread use of these vaccines in controlling the outbreak. Although most cases of C19-VAL were diagnosed histologically as reactive or negative for malignancy, other diagnoses including florid lymphoid hyperplasia, KFD, and granulomatous inflammation have been reported. Metastases can occur in patients with a history of malignancy who have been recently vaccinated, and the lymphadenopathy in these cases is likely related to the underlying malignancy. Awareness of these pathological findings, along with their associated clinical and radiological findings, may help to guide the management of this population of patients.

## Figures and Tables

**Figure 1 jcm-11-06290-f001:**
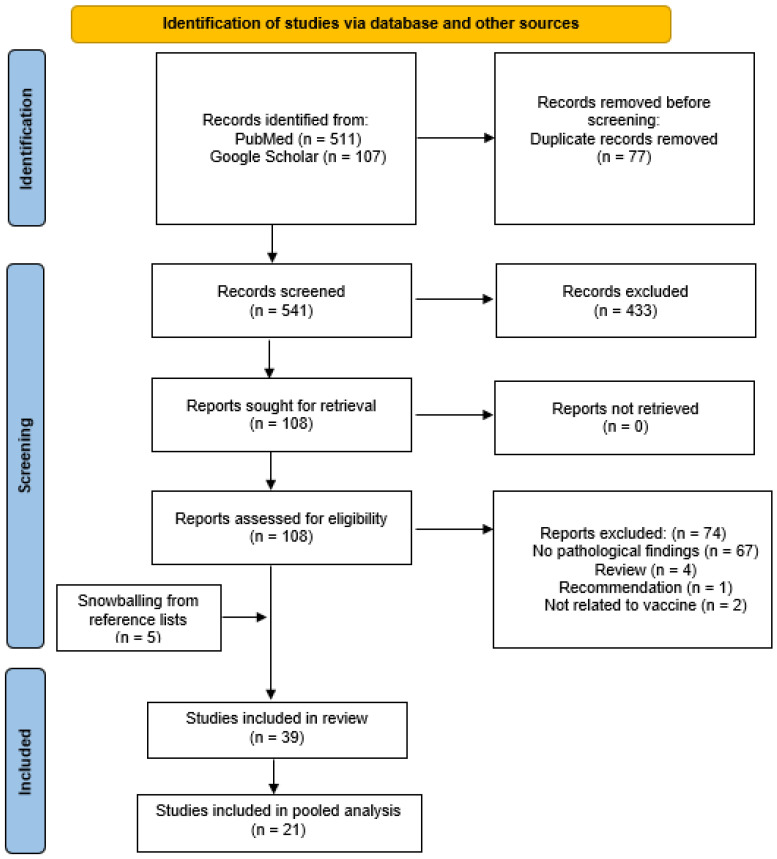
PRISMA diagram.

**Table 1 jcm-11-06290-t001:** Pooled analysis. CL, clinical lymphadenopathy; SLDI, subclinical lymphadenopathy detected on imaging; SD, standard deviation; N, number of; NET, neuroendocrine tumor; RCC, renal cell carcinoma; US, ultrasound; PET/CT, positron emission tomography/computed tomography; MRI, magnetic resonance imaging; KFD, Kikuchi–Fujimoto Disease; FNA, fine-needle aspiration.

	Total	CL				SLDI
		Overall	Reactive Changes or Negative for Malignancy	Florid Lymphoid Hyperplasia	Kikuchi–Fujimoto Disease	Overall
Number of patients	37	18	13	2	3	19
Mean age ± SD (years)	47.8 ± 19.1	37.8 ± 15.6	44.2 ± 13.1	18.0 ± 5.0 *	23.3 ± 7.5 *	57.2 ± 17.3 **
Number of females (%)	23 (62.2)	9 (50.0)	7 (53.8)	1 (50.0)	1 (33.3)	14 (73.7)
Medical history, *n* (%)						
No history	9 (24.3)	9 (50.0)	7 (53.8)	1 (50.0)	1 (33.3)	0 (0.0)
Family history of breast cancer	2 (5.4)	2 (11.1)	2 (11.1)	0 (0.0)	0 (0.0)	0 (0.0)
Personal history of breast cancer	7 (18.9)	0 (0.0)	0 (0.0)	0 (0.0)	0 (0.0)	7 (36.8)
Personal history of lung cancer	2 (5.4)	2 (11.1)	2 (11.1)	0 (0.0)	0 (0.0)	0 (0.0)
Personal history of cecum/appendix NET	2 (5.4)	1 (5.5)	1 (5.5)	0 (0.0)	0 (0.0)	1 (5.3)
Personal history of melanoma	8 (21.6)	0 (0.0)	0 (0.0)	0 (0.0)	0 (0.0)	8 (42.1)
Personal history of Merkel cell carcinoma	2 (5.4)	0 (0.0)	0 (0.0)	0 (0.0)	0 (0.0)	2 (10.5)
Personal history of RCC	1 (2.7)	0 (0.0)	0 (0.0)	0 (0.0)	0 (0.0)	1 (5.3)
Non-neoplastic/malignant history	3 (8.1)	3 (16.7)	0 (0.0)	1 (50.0)	2 (66.6)	0 (0.0)
Not reported	1 (2.7)	1 (5.5)	1 (7.7)	0 (0.0	0 (0.0)	0 (0.0)
**Dose and type of last vaccine, *n* (%)**						
1st dose of Pfizer-Bio-Ntech	13 (35.1)	7 (38.9)	6 (46.2)	0 (0.0)	1 (33.3)	6 (31.6)
2nd dose of Pfizer-Bio-Ntech	7 (18.9)	1 (5.5)	1 (7.7)	0 (0.0)	0 (0.0)	6 (31.6)
Unspecified dose of Pfizer-Bio-Ntech	1 (2.7)	0 (0.0)	0 (0.0)	0 (0.0)	0 (0.0)	1 (5.3)
1st dose of Moderna	3 (8.1)	2 (11.1)	2 (15.4)	0 (0.0)	0 (0.0)	1 (5.3)
2nd dose of Moderna	3 (8.1)	2 (11.1)	1 (7.7)	1 (50.0)	0 (0.0)	1 (5.3)
Unspecified dose of Moderna	1 (2.7)	1 (5.5)	1 (7.7)	0 (0.0)	0 (0.0)	0 (0.0)
1st dose of AstraZeneca	1 (2.7)	1 (5.5)	1 (7.7)	0 (0.0)	0 (0.0)	0 (0.0)
2nd dose of AstraZeneca	1 (2.7)	0 (0.0)	0 (0.0)	0 (0.0)	0 (0.0)	1 (5.3)
1st dose of unspecified vaccine	3 (8.1)	3 (16.7)	1 (7.7)	0 (0.0)	2 (66.6)	0 (0.0)
1st dose of Vaxzevria	2 (5.4)	0 (0.0)	0 (0.0)	0 (0.0)	0 (0.0)	2 (10.5)
2nd dose of CureVac	1 (2.7)	0 (0.0)	0 (0.0)	0 (0.0)	0 (0.0)	1 (5.3)
Not reported	1 (2.7)	1 (5.5)	0 (0.0)	1 (50.0)	0 (0.0)	0 (0.0)
Duration from last vaccination to CL or SLDI, mean ± SD (days)	14.5 ± 11.0	12.5 ± 7.9	10.9 ± 6.3	10.5 ± 3.5	20.7 ± 10.5 *	16.5 ± 12.9
**Laterality of lymphadenopathy compared with site of vaccination, *n* (%)**						
Ipsilateral	23 (62.2)	10 (55.6)	9 (69.2)	1 (50.0)	0 (0.0)	13 (68.4)
Contralateral	2 (5.4)	2 (11.1)	2 (15.4)	0 (0.0)	0 (0.0)	0 (0.0)
Not reported	12 (32.4)	6 (33.3)	2 (15.4)	1 (50.0)	3 (100.0)	6 (31.6)
Site of lymphadenopathy, *n* (%)	**out of 32**	**out of 13**	**out of 8**			
Cervical	4 (12.5)	3 (23.1)	2 (25.0)	0 (0.0)	1 (33.3)	1 (5.3)
Axilla	18 (56.3)	4 (30.8)	1 (12.5)	1 (50.0)	2 (66.6)	14 (73.7) **
Supraclavicular	7 (21.9)	6 (46.2)	5 (62.5)	1 (50.0)	0 (0.0)	1 (5.3) **
Others	2 (6.3)	1 (7.7)	0 (0.0)	0 (0.0)	0 (0.0)	1 (5.3)
Not reported	5 (15.6)	0 (0.0)	0 (0.0)	0 (0.0)	0 (0.0)	5 (26.3)
**Additional clinical symptoms aside from lymphadenopathy, *n* (%)**						
No other symptoms	2 (5.4)	2 (11.1)	2 (15.4)	0 (0.0)	0 (0.0)	0 (0.0)
Pain	4 (10.8)	4 (30.8)	3 (23.1)	1 (50.0)	0 (0.0)	0 (0.0)
Fever	6 (16.2)	6 (46.2)	2 (15.4)	1 (50.0)	3 (100.0)	0 (0.0)
Fatigue/malaise	2 (5.4)	2 (11.1)	1 (7.7)	1 (50.0)	0 (0.0)	0 (0.0)
Myalgia	2 (5.4)	2 (11.1)	2 (15.4)	0 (0.0)	0 (0.0)	0 (0.0)
Dysphagia	1 (2.7)	1 (5.5)	1 (7.7)	0 (0.0)	0 (0.0)	0 (0.0)
Chills	1 (2.7)	1 (5.5)	1 (7.7)	0 (0.0)	0 (0.0)	0 (0.0)
Others	2 (5.4)	2 (11.1)	1 (7.7)	1 (50.0)	0 (0.0)	0 (0.0)
Not reported	24 (64.9)	5 (27.8)	5 (38.5)	0 (0.0)	0 (0.0)	19 (100.0)
Largest dimension of lymph node, mean ± SD (mm)	20.8 ± 13.3	21.1 ± 14.7	22.1 ± 18.2	15.5 ± 5.5	22.3 ± 7.1	19.7 ± 2.9
**Imaging modality with abnormal findings, *n* (%)**						
US	11 (29.7)	7 (38.9)	7 (53.8)	0 (0.0)	0 (0.0)	4 (21.1)
PET/CT	8 (21.6)	2 (11.1)	2 (15.4)	0 (0.0)	0 (0.0)	6 (31.6)
CT/MRI	11 (29.7)	8 (44.4)	4 (30.8)	1 (50.0)	3 (100.0)	3 (15.8)
Not reported	10 (27.0)	2 (11.1)	1 (7.7)	1 (50.0)	0 (0.0)	8 (42.1)
**Indication for aspiration or biopsy, *n* (%)**						
Suspicion of malignancy	15 (40.5)	8 (44.4)	7 (53.8)	1 (50.0)	0 (0.0)	7 (36.8)
Family history of malignancy	1 (2.7)	1 (5.5)	1 (7.7)	0 (0.0)	0 (0.0)	0 (0.0)
Palpable mass	2 (5.4)	2 (11.1)	2 (15.4)	0 (0.0)	0 (0.0)	0 (0.0)
Suspicion of lymphadenitis and/or KFD	2 (5.4)	2 (11.1)	2 (15.4)	0 (0.0)	0 (0.0)	0 (0.0)
Patient’s preference	2 (5.4)	0 (0.0)	0 (0.0)	0 (0.0)	0 (0.0)	2 (10.5)
Oncologic management	1 (2.7)	0 (0.0)	0 (0.0)	0 (0.0)	0 (0.0)	1 (5.3)
Not reported	15 (40.5)	6 (46.2)	2 (15.4)	1 (50.0)	3 (100.0)	9 (47.4)
Type of procedure, *n* (%)						
FNA	9 (24.3)	7 (38.9)	7 (53.8)	0 (0.0)	0 (0.0)	2 (10.5)
Core needle biopsy	12 (32.4)	7 (38.9)	5 (38.5)	0 (0.0)	2 (66.6)	5 (26.3)
Excision biopsy	12 (32.4)	5 (27.8)	2 (15.4)	2 (100.0)	1 (33.3)	7 (36.8)
Others	4 (10.8)	0 (0.0)	0 (0.0)	0 (0.0)	0 (0.0)	4 (21.1)
Not reported	1 (2.7)	0 (0.0)	0 (0.0)	0 (0.0)	0 (0.0)	1 (5.3)
Pathological diagnosis, *n* (%)						
Reactive/negative for malignancy	28 (75.7)	13 (72.2)	-	-	-	15 (78.9)
Florid lymphoid hyperplasia	2 (5.4)	2 (11.1)	-	-	-	0 (0.0)
Kikuchi–Fujimoto Disease	3 (8.1)	3 (16.7)	-	-	-	0 (0.0)
Granulomatous inflammation	2 (5.4)	0 (0.0)	-	-	-	2 (10.5)
Metastatic malignancy	2 (5.4)	0 (0.0)	-	-	-	2 (10.5)

* Statistically significant compared with reactive changes or negative for malignancy; ** statistically significant compared with overall clinical lymphadenopathy.

**Table 2 jcm-11-06290-t002:** Findings of clinical lymphadenopathy (CL) studies. NR, not reported; F, female; M, male; US, ultrasound; PET/CT, positron emission tomography/computed tomography; MRI, magnetic resonance imaging; FNA, fine-needle aspiration.

Author (Country) [Type of Publication]	Number of Patients with Pathological Findings	Age (Sex)	Significant History	Type and Dose of Most Recent Vaccine	Site of Vaccine	Lymphadenopathy (Palpable or Painful)	Other Clinical Symptoms	Duration from Last Vaccination to Lymphadenopathy (Days)	Laterality of Lymphadenopathy Compared with Site of Vaccination	Site of Lymphadenopathy	Largest Dimension of Lymph Node (mm)	Ultrasound Finding(s)	PET/CT/MRI Finding(s)	Indication for Aspiration or Biopsy	Type of Pathological Specimen	Pathological Finding(s)	Management	Outcome
Cardoso et al. (Portugal) [Case report]	1	48 (F)	Family history of breast cancer	1st dose of Pfizer-BioNTech (2nd dose given after lymphadenopathy)	NR	Palpable	No other systemic symptoms	14	NR	Posterior edge of lower third of right sternocleidomastoid muscle	14	Increase in echogenicity and sphericity index without a defined hilum	CT: right lateral cervical adenopathies	Family history of malignancy	FNA and excision biopsy	FNA: atypical lymphoid cytology Biopsy: reactive follicular hyperplasia	NR	NR
Faermann et al. (Israel) [Retrospective]	7	NR (F)	Breast cancer, family history, BRCA carrier	Pfizer-Bio-Ntech	NR	NR	NR	NR	Ipsilateral	Axilla	NR	NR	NR	Clinical suspicion of metastases	US-guided core needle biopsy	Reactive	NR	NR
Felices-Farias et al. (Spain) [NR]	11	NR (NR)	NR	Pfizer-Bio-Ntech	Arm	Painful	NR	NR	Ipsilateral	Axilla and supraclavicular	NR	NR	NR	Suspicion of malignancy	US-guided core needle biopsy	Reactive paracortical/interfollicular hyperplasia	NR	NR
Fernandez-Prada et al. (Spain) [Case series]	5	NR (NR)	NR	Pfizer-Bio-Ntech and Moderna	NR	Palpable	Pain, swelling	1 to 9	Ipsilateral	Supraclavicular	NR	NR	NR	NR	Aspiration	Reactive; lymphocytic infiltrate and active germinal centers	NR	NR
Ganga et al. (USA) [Case report]	1	58 (M)	Nil	Moderna	NR	Palpable	Fever, fatigue, myalgia, dysphagia	2	NR	Left neck	59	Mass effect upon left internal jugular vein	CT: marked irregular thickening and inflammation of sternocleidomastoid area;	NR	US-guided biopsy	Negative for malignancy	Empiric antibiotics, dexamethasone	Significant reduction in swelling on 2nd day of admission; no residual symptoms 2 weeks later
Hagen et al. (Switzerland) [Case series]	5	66 (M)	Lung cancer	2nd dose of Moderna	Left arm	Palpable	NR	22	Ipsilateral	Cervical level IV, supra-, infra-, or retroclavicular, axilla	10 to 24	Ovoid to rounded shapes, sharp borders, partially detectable hilum; some with suspicious findings such as spherical shape with loss of hilum	PET/CT: enlarged and very highly FDG-active axillary lymph nodes	Clinical suspicion of metastases	FNA	Reactive; no evidence of metastases	NR	Complete regression at 2-month follow-up
		41 (F)	Nil	1st dose of Moderna	Left arm	Palpable	NR	3	Ipsilateral				NR	Palpable mass	FNA	Reactive; no evidence of metastases	NR	
		47 (F)	Nil	1st dose of Pfizer-BioNTech	Left arm	Palpable	NR	19	Ipsilateral				NR	Palpable mass	FNA	Reactive; no evidence of metastases	NR	
		47 (F)	Appendix NET	1st dose of Moderna	Left arm	Palpable	NR	8	Ipsilateral				NR	Clinical suspicion of metastases	FNA	Reactive; no evidence of metastases	NR	
		52 (M)	Lung cancer	2nd dose of Pfizer-BioNTech	Right arm	Palpable	NR	12	Contralateral				PET/CT: moderate and very high FDG activity	Clinical suspicion of metastases	FNA	No evidence of metastases or lymphoma	NR	
Kado et al. (Japan) [Case report]	1	31 (F)	NR	1st dose of Pfizer-Bio-Ntech	Left arm	Palpable	Pain	8	Ipsilateral	Left-upper clavicle, left-lower scapula	15	Rounded and fatty hilum not observed in some lymph nodes	CT: deep cervical lymphadenopathies on the left side	Suspicion of malignancy	Needle biopsy	Follicular hyperplasia, no evidence of malignancy	NR	Decrease in size and number of unilateral lymphadenopathies after 6 weeks; impalpable subcutaneous nodules after 3 months
Kim et al. (South Korea) [Case report]	1	36 (F)	Nil	1st dose of Pfizer-Bio-Ntech	Left arm	Palpable, pain	Discomfort, swelling, pain	17	Ipsilateral	Left supraclavicular, level V	7	More than 5 lymph nodes, round, thickened cortex, loss of normal fatty hilum, ill-defined border with perinodal fat hyper echogenicity	NR	Possibility of lymphadenitis and Kikuchi disease	US-guided core needle biopsy	Reactive; predominantly small mature T-lymphocytes with small mature B-lymphocytes; negative EBV-encoded small RNA	Symptom relief; advised for second dose of vaccine in the contralateral arm	Reduction in size and extent of lymphadenopathy; newly developed palpable lesions in the right supraclavicular region after 2nd dose of vaccine in the right arm; lymphadenopathy eventually subsided
Larkin et al. (USA) [Case series]	1	16 (M)	Nil	1st dose	Left arm	Palpable	No other symptoms	14	Ipsilateral	Left supraclavicular	10	NR	NR	Suspicion of malignancy	Excision biopsy	Reactive follicular hyperplasia and focal increased EBV-positive cells (serum negative EBV PCR, EBER-ISH suggestive of prior infection, EBV IgG positive)	NR	NR
Ozutemiz et al. (USA) [Case series]	1	38 (F)	Family history of breast cancer	1st dose of Pfizer-BioNTech	Left arm	Pain	Axillary pain	8	Ipsilateral	Left axilla	NR	Cortical thickness of 6 mm	NR	NR	US-guided core needle biopsy	Reactive follicular hyperplasia; no evidence of malignancy	NR	NR
Park et al. (South Korea) [Case report]	1	61 (M)	Nil	1st dose of AstraZeneca	Left arm	Palpable	Fever, chills, muscle pain	14	Contralateral	Right supraclavicular	40	NR	CT: enlarged, clusters and conglomerated lymph nodes with perinodal infiltration in the right supraclavicular area; no necrosis	Clinical suspicion of malignancy and Kikuchi disease	US-guided core needle biopsy	Reactive hyperplasia with capsular and trabecular fibrosis; negative for malignancy	NR	Improvement of lymphadenopathy
Tan et al. (Singapore) [Case report]	1	34 (M)	Nil	1st dose of Pfizer-BioNTech	Left arm	Palpable, pain	Soreness over vaccination site	1	Ipsilateral	Left supraclavicular	10	Minimal internal vascularity and no calcification. Hilum was not clearly visualized; however, no sonographically suspicious features	NR	To exclude an occult metastatic malignancy	FNA	Reactive follicular hyperplasia	NR	Complete resolution of lymphadenopathy
Larkin et al. (USA) [Case series]	1	13 (M)	Nil	NR	Left arm	Palpable	Transiently painful	14	Ipsilateral	Left supraclavicular	10	NR	NR	Suspicion of malignancy	Excision biopsy	Florid, reactive follicular hyperplasia, with foci of follicular lysis, increased immunoblasts and progressive transformation of germinal centers	NR	NR
Tintle et al. (USA) [Case report]	1	23 (F)	Asthma, eczema, hypothyroidism	2nd dose of Moderna	NR	Palpable	Fever, malaise, vomiting, acute kidney injury	7	NR	Left axilla and abdomen	21	NR	CT: left axillary lymphadenopathy and multiple enlarged lymph nodes in the abdomen	NR	Excision biopsy	Florid follicular and interfollicular lymphoid and Langerhans cell hyperplasia	Dexamethasone and anakinra	Recovered within 2 weeks
Soub et al. (Qatar) [Case report]	1	18 (M)	Steroid dependent minimal change renal disease	1st dose of Pfizer-BioNTech	NR	Palpable	Fever	10	NR	Left neck	15	NR	CT: multiple left cervical and axillary lymph nodes, with the largest one in left supraclavicular region measuring 11 × 10 mm	NR	Excision biopsy	Kikuchi–Fujimoto disease	NR	Discharged in good condition
Tan et al. (Singapore) [Case series]	2	18 (F)	Nil	1st dose	NR	NR	Fever	35	NR	Left axilla	20	NR	CT: enlarged left supraclavicular, subpectoral and axillary lymph nodes	NR	US-guided core biopsy	Kikuchi–Fujimoto disease	Symptom relief	Resolution of symptoms by day 58
		34 (M)	Diabetes mellitus, hypertension	1st dose	NR	NR	Fever	17	NR	Left axilla	32	NR	CT: enlarged left axillary lymph nodes	NR	US-guided core biopsy	Kikuchi–Fujimoto disease	Symptom relief	Resolution of symptoms by day 38

**Table 3 jcm-11-06290-t003:** Findings of subclinical lymphadenopathy detected on imaging (SLDI) studies. NR, not reported; F, female; M, male; US, ultrasound; PET/CT, positron emission tomography/computed tomography; MRI, magnetic resonance imaging; FNA, fine-needle aspiration.

Author (Country) [Type of Publication]	Number of Patients with Pathological Findings	Age (Sex)	Significant History	Type and Dose of Most Recent Vaccine	Site of Vaccine	Duration from Last Vaccination to Development of Lymphadenopathy (Days)	Laterality of Lymphadenopathy Compared with Site of Vaccination	Site of Lymphadenopathy	Largest Dimension of Lymph Node (mm)	Ultrasound Finding(s)	PET/CT/MRI Finding(s)	Indication for Aspiration or Biopsy	Site of Aspiration or Biopsy	Type of Pathological Specimen	Pathological Finding(s)
Aalberg et al. (USA) [Case report]	1	74 (M)	Stage IV clear cell RCC with bone and lung metasatses	2nd dose of Moderna	Left deltoid	2	Ipsilateral	Left axilla	23 × 12 mm	NR	Standardized uptake value of 9.7, additional sub-centimeter left axillary lymph nodes with maximum SUV of 4.1	Known history of metastatic malignancy and variable PET findings to known metastatic lesions in lung	Left axilla	FNA	Polymorphous lymphoid population compatible with reactive lymph node; negative for metastatic carcinoma
Ashoor et al. (UK) [Case series]	1	61 (F)	High-grade DCIS	2nd dose of AstraZeneca	Left arm	1	Ipsilateral	Left axilla	NR	Thickened cortex > 3 mm, intact fatty hilum; eccentrically thickened cortex that measured 4.9 mm in one lymph node	NR	Indeterminate US finding and extensive malignant-appearing calcification on screening mammography	Left axilla	Biopsy	Benign reactive changes
Eifer et al. (Israel) [Retrospective]	1	41 (F)	Breast cancer	2nd dose of Pfizer-BioNTech	Both arms	1	Ipsilateral	Left axilla	NR	NR	PET/CT: Increased FDG uptake	Clinical suspicion of metastases	Left axilla	US-guided core needle biopsy	Reactive; no evidence of metastases
Lane et al. (USA) [Case series]	1	44 (F)	Left-breast high-grade DCIS	1st dose of Pfizer-Bio-Ntech	Left arm	4	Ipsilateral	Left axilla	NR	NR	MRI: asymmetrical left axillary nodes compared with the right	Oncologic management	Left axilla	US-guided biopsy	Benign; no metastases
Lim et al. (South Korea) [Case series]	3	61 (F)	Right-breast IDC with ipsilateral axillary nodal metastases	1st dose of Vaxzevria	Left arm	22	Ipsilateral	Left axilla	NR	NR	NR	Patient’s preference	Left axilla	US-guided 14-gauge gun biopsy	Benign hyperplasia
		75 (F)	Right-breast IDC without axillary nodal metastases	2nd dose of Pfizer-BioNTech	NR	19	NR	Left axilla	NR	Maximum cortical thickness of 5.38 mm	CT/MRI: mean length/width ratio less than 1.5	NR	Left axilla	NR	Reactive hyperplasia
		71 (F)	Right-breast IDC without axillary nodal metastases	1st dose of Vaxzevria	Left arm	14	Ipsilateral	Left axilla	NR	Smooth and diffuse enlargement and borderline maximum cortical thickness of 3 mm	CT: length/width ratio less than 1.5 and interval change in maximum cortical thickening greater than 2 times compared with previous scans	NR	Left axilla	US-guided biopsy	Benign hyperplasia
Ozutemiz et al. (USA) [Case series]	1	46 (F)	Left-breast IDC	1st dose of Pfizer-BioNTech	Left arm	15	Ipsilateral	Left axilla	20 × 12 mm	NR	PET/CT: supraclavicular lymph nodes; multiple enlarged hypermetabolic lymph nodes in left axilla, largest 20 × 12 mm	Patient’s preference	Left axilla and supraclavicular	US-guided core needle biopsy	Benign and reactive; no evidence of breast cancer metastases
Robinson et al. (USA) [Retrospective]	1	NR (F)	Breast cancer	NR	NR	NR	NR	NR	NR	NR	NR	NR	NR	Biopsy	Negative for malignancy
Prieto et al. (USA) [Case report]	1	48 (F)	Melanoma	1st dose of Moderna	Right arm	5	Ipsilateral	Right axilla	NR	NR	PET/CT: substantial FDG avidity in right axilla and neck	Suspicion of malignancy	Right axilla	Image-guided biopsy	Reactive lymphoid tissue; negative for metastatic melanoma
Trikannad et al. (NR) [Case report]	1	57 (F)	Melanoma	Pfizer-BioNTech	Right arm	21	Ipsilateral	Mediastinum, right axilla and right neck	NR	NR	PET/CT: increased uptake	Suspicion of malignancy	Mediastinum	FNA	Non-caseating granulomas and reactive changes; no evidence of malignancy
Placke et al. (Germany) [Case series]	8	28 (F)	Melanoma	1st dose of Pfizer-BioNTech	Left arm	28	Ipsilateral in three patients	Left axilla in three patients	16 mm	Echo-deficient, increased marginal vascularisation	NR	Suspicion of malignancy	Left axilla	Selective lymph node excision	Marked lymphofollicular hyperplasia
		43 (F)	Melanoma	2nd dose of Pfizer-BioNTech	Left arm	50			NR	NR	NR	Suspicion of malignancy	Left axilla	Complete lymphadenectomy	Sarcoid-like reaction; no evidence of melanoma metastases
		54 (F)	Melanoma	2nd dose of CureVac	Left arm	30			NR	NR	NR	NR	NR	Sentinel lymph node excision	Metastases in 2 patients
		58 (M)	Merkel cell carcinoma	1st dose of Pfizer-BioNTech	Left arm	7			NR	NR	NR	NR	NR	Selective lymph node excision	
		77 (M)	Melanoma	1st dose of Pfizer-BioNTech	Left arm	11			NR	NR	NR	NR	NR	Sentinel lymph node excision	
		91 (M)	Merkel cell carcinoma	2nd dose of Pfizer-BioNTech	Left arm	16			NR	NR	NR	NR	NR	Sentinel lymph node excision	
		44 (M)	Melanoma	1st dose of Pfizer-BioNTech	Left arm	15			NR	NR	NR	NR	NR	Sentinel lymph node excision	
		84 (F)	Melanoma	2nd dose of Pfizer-BioNTech	Left arm	12			NR	NR	NR	NR	NR	Sentinel lymph node excision	
Pudis et al. (Spain) [Case report]	1	30 (F)	Cecum-appendix NET	2nd dose of Pfizer-BioNTech	Right arm	40	Ipsilateral	Right axilla and supraclavicular	NR	Normal morphology, central hilum, thin cortex	PET/CT: intense uptake in right axillary and supraclavicular region	NR	Right axilla	Surgical resection	Benign reactive changes

**Table 4 jcm-11-06290-t004:** Findings from non-COVID-19 vaccine-associated lymphadenopathy. BCG, bacillus Calmette–Guérin; HPV, human papillomavirus; JEV, Japanese encephalitis virus.

Author (Year and Country of Publication)	Type of Publication	Type of Vaccine	Key Findings
Aelami et al. (2015, Iran)	Retrospective	BCG	A total of 12/13 (92.3%) biopsies or aspirations of distant lymph nodes showed presence of granuloma and/or acid-fast bacilli.
Barouni et al. (2003, Brazil)	Case report	BCG	A 2-year-old male with BCG vaccination at 1 month of age developed lymphadenopathy secondary to atypical tuberculosis. Lymph node aspirates showed presence of atypical tuberculosis.
Biers et al. (2007, UK)	Case report	BCG	A 64-year-old male that received intravesical BCG for urothelial carcinoma developed solitary iliac lymphadenopathy. Biopsy showed granulomatous inflammation.
Gupta et al. (1996, India)	Retrospective	BCG	A total of 112 cases of BCG lymphadenitis with fine-needle aspiration that showed caseating granulomatous inflammation.
Pal et al. (2015, India)	Prospective	BCG	A total of 30 cases of BCG lymphadenitis with needle aspiration smears showing caseating granulomatous inflammation and reactive hyperplasia.
Wang et al. (2019, Taiwan)	Case report	BCG	A 2-year-old female with left axillary lymphadenopathy and caseating granulomatous inflammation seen on excision biopsy.
Dotlic et al. (2012, Croatia)	Case report	BCG and Hepatitis B	A 2-week-old male with inguinal lymphadenopathy after receiving both vaccines at birth. Excision biopsy showed caseating granulomatous inflammation and hyperimmune post-vaccinal reaction involving an atypical T cell proliferation.
Toy et al. (2010, Turkey)	Case report	H1N1	A 23-year-old male with a left supraclavicular painful lymphadenopathy received H1N1 vaccination one week prior to admission. The lymph node was resected and showed post-vaccinal lymphadenitis with CD30-positive immunoblasts, as well as unusually large immunoblasts resembling Hodgkin cells.
Pereira et al. (2019, Portugal)	Case series	HPV 9-valent	An 11-year-old male with inferior cervical and supraclavicular lymphadenopathy that received HPV 9-valent vaccine in the previous week. He subsequently underwent biopsy of the two largest lymph nodes, that showed follicular lymphoid hyperplasia and parafollicular hyperplasia.
Watanabe et al. (2012, Japan)	Case report	HPV and JEV	A 14-year-old female with bilateral tender cervical lymphadenopathy who received HPV and JEV vaccine three days prior. Excision biopsy showed Kikuchi–Fujimoto disease.
Dorfman et al. (1966, USA)	Case report	Measles	An 11-month-old female with left inguinal lymphadenopathy that received a measles vaccine 13 days prior. The resected lymph nodes showed lymphoid hyperplasia.
Sumaya et al. (1976, USA)	Case report	Rubella	A 6-year-old male with painless cervical lymphadenopathy that enlarged after he was vaccinated with HPV-77 DK/5 rubella vaccine. Biopsies showed sinus histiocytosis with massive lymphadenopathy.
White et al. (2012, USA)	Case report	Tetanus	A 50-year-old female with left supraclavicular lymphadenopathy 72 h after she was given the booster dose of tetanus toxoid. Excision biopsy showed sheets of small-to-medium-sized atypical lymphocytes with a flow cytometry analysis interpretation of atypical T cell population co-expressing CD4 and CD8. This was initially reported as a T cell lymphoproliferative disorder. However, after a review, this was reported as ‘pseudolymphomatous’ florid proliferation of CD4 T cells in response to the vaccine.
Hartsock et al. (1967, USA)	Retrospective	Smallpox, cholera, typhus, tetanus, diphtheria, pertussis, Salk (polio), influenza	A total of 20 cases of post-vaccinal lymphadenitis, with 9 cases initially diagnosed as lymphoma. All 20 cases showed a diffuse, follicular, or combined diffuse and follicular hyperplasia, an increased number of reticular lymphoblasts, vascular and sinusoidal changes, and mixed inflammatory response with a variable number of eosinophils, plasma cells, and mast cells.

## Data Availability

All data presented are available within the manuscript.

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
