# Peer review of "Pathological Findings in COVID-19 and Non-COVID-19 Vaccine-Associated Lymphadenopathy: A Systematic Review"

_jcm, 2022, doi:10.3390/jcm11216290_

Round 1

Reviewer 1 Report (New Reviewer)

Despite the  value of the topic, the clinical significance of this study is clearly limited by the sample size.

Nevertheless the methodology is correct and results well described. I belive that, once the limitation have been made clear to readers, the study deserves pubblication as it is, for the intrinsic significance of its objective. 

Author Response

Reviewer 1

Despite the value of the topic, the clinical significance of this study is clearly limited by the sample size.

Nevertheless the methodology is correct and results well described. I belive that, once the limitation have been made clear to readers, the study deserves pubblication as it is, for the intrinsic significance of its objective.

Authors’ response:

We thank the reviewer for the comments. The limitation of a small sample size was mentioned in the original version of the manuscript, in page 11, line 17.

Reviewer 2 Report (New Reviewer)

Thank you very much for inviting me to review this interesting paper. This are my suggestions:

ABSTRACT: line 2: widespread use in controlling the outbreak - add "of the vaccine" after the word use.

Add absolute number in the results section of the abstract. Please add a conclusion, you are only repeating the results section.

INTRODUCTION: line 7, the comma after (mRNA) in not in place, should after the word vaccine.

Paragraph 3: You wrote:   Although some of these radiological findings may favour a reactive lymphadenopathy, studies have also reported suspicious ultrasound findings in lymphadenopathy after COVID-19 vaccination in patients being followed up (ADD HERE "FOR") skin cancer, raising a diagnostic conundrum between a reactive and malignant process (A PERIOD IS MISSING) - YOU SHOULD MENTION ALSO BREAST CANCER, FOLLOW UP BUT MOSTLY SCREENING FOR BREAST CANCER, WHICH INVOLVES A GREAT NUMBER OF HEATHY WOMEN - ADD REFERENCES FOR THIS.

Last paragraph of introduction: Histopathological and cytopathological findings, obtained through procedures including fine needle aspiration and biopsy (ADD "CORE NEEDLE" BEFORE THE WORD BIOPSY)

MATERIALS AND METHODS:

First paragraph: 

"A literature (I THINK THE WORD "SEARCH" IS MISSING HERE) was performed, in accordance with the PRISMA statement,19 through PubMed and Google Scholar, on 11 December 2021."

"The reference lists of full-text articles were search for relevant studies that were not identified in the initial search." - PLEASE IMPROVE THIS SENTENCE, IT IS HARD TO UNDERSTAND.

Third paragraph: We extracted the pertinent details from each study, and these include: I THINK IT SHOULD BE "INCLUDED"

Last paragraph of materials and methods: "SLDI patients are (ADD THE WORD "DETECTED") on routine imaging follow-up for conditions including a history of malignancy." AGAIN, ADD HERE SCREENING (SINCE SCREENING FOR BREAST CANCER ACCOUNTS FOR A GREAT NUMBER OF THESE PATIENTS - ULTRASOUND PERFORMED AS SCREENING FOR BREAST CANCER INCLUDES THE AXILLAE.) JUST MENTION "CONDITIONS INCLUDING A HISTORY OF MALIGNANCY, OR SCREENING FOR CANCER' SUCH AS BREAST CANCER.

RESULTS:

Second paragraph: Most studies are case reports (n = 12, 48.0%), followed by case series (n = 9, 36.0%) (A COMMA IS MISSING HERE) retrospective studies (n = 3, 12.0%), and not reported in one study (4.0%).

Third paragraph: Pooled analysis of 21 studies included 37 patients, with a mean age of 47.8 ± 19.1 years (ADD THE WORD "OLD" HERE) and 23 (62.2%) (ADD THE WORD "PATIENTS") were females.

ADD ABSOLUTE NUMBER AND NOT ONLY PERCENTAGES WHERE IT IS MISSING.

Fifth paragraph: Last sentence - The other procedures that were performed include surgical resection and complete lymphadenectomy - I THINK IT SHOULD BE "INCLUDED"

Sixth paragraph: Most cases of lymphadenopathy that were sent for pathological examination, were diagnosed as reactive or negative for malignancy (75.5%) (ADD A COMMA HERE) while florid lymphoid hyperplasia was reported in 5.4%.

Seventh paragraph: A total of 15 studies reported pathological findings of COVD-19 vaccine-associated CL. (YOU HAVE A TYPO IN COVID-19

YOU INCLUDED STUDY REFERENCE NUMBER 22 AS CLINICAL LYMPHADENOPATHY, BUT THIS IS SUBCLINICAL LYMPHADENOPATHY (IT IS CLEARLY WRITTEN THAT PATIENTS THAT HAD A BIOSPY HAD THEIR LYMPHADENOPATHY DETECTED ON IMAGING, WHICH MEANS THIS IS SUBCLINICAL). THIS STUDY SHOULD BE IN THE GROUP ANALYSIS OF SUBCLINICAL LYMPHADENOPATHY - PLEASE CORRECT THIS, AND REVIEW THE OTHER STUDIES THAT YOU INCLUDED IN EACH SUBGROUP TO ENSURE THEY ARE IN THE RIGHT SUBGROUP. PLEASE ADD IN THE TABLES IN EACH STUDY WHAT WAS THE CLINICAL FINDING THAT LED YOU TO CLASSIFY IT AS CLINICAL LYMPHADENOPATHY (CLINICAL SHOULD BE PALPABLE LYMPH NODES OR PAINFUL LYMPH NODES). IF LYMPHADENOPATHY WAS DETECTED ON IMAGING, THIS SHOULD BE CONSIDERED SUBCLINICAL. REVIEW ALL STUDIES FOR THAT PLEASE. IN ADDITION, YOU HAVE IN THAT STUDY US FINDINGS AND LARGEST DIMENSION OF LN (YOU WROTE AS NR), SO REVIEW AS WELL AND ALSO IN OTHER STUDIES.

DISCUSSION:

Third paragraph: For (CHANGE FOR "AMONG PATIENTS") patients with SLDI, although most cases were diagnosed as reactive or negative for malignancy, granulomatous inflammation and metastases were also reported in a proportion of this population. HOW MANY PATIENTS HAD METASTASIS? ADD HERE HOW MANY AND IF ONLY ONE, ADD THAT IT WAS RARE AND ALTHOUGH RARE, YTHIS DD SHOULD BE CONSIDERED AS YOU MENTIONED.

Third paragraph: . Despite these associations of COVID-19 vaccine with lymphadenopathy that were diagnosed as KFD, granulomatous inflammation and metastases, it remains unclear whether these conditions are related or unrelated to the vaccine. The etiology and pathogenesis of KFD remains unclear, with viruses being postulated to be a key inciting agent. As such, based on findings from this study, we can only conclude the association of COVID-19 vaccine with these lymphadenopathies. PLEASE DON'T SAY THAT VACCINATION IS ASSOCIATED WITH LYMPH NODE METASTASIS!! YOU CANNOT CONCLUDE THIS AND IT IS NOT ASSOCIATED WITH THE VACCINE, BUT WITH THE UBDERLYING CANCER! REMOVE THIS ASSOCIATION AND REMOVE THIS LAST SENTENCE.

Sixth paragraph: Most of the cases WERE

Seventh paragraph: second sentence is too long, please split it, and expain that effectiveness of vaccine was to REDUCE these outcomes.

With the likely need for future booster doses, lymphadenopathy after the third COVID19 vaccine has also already been reported6 and is expected to increase with time. - WHY ALSO? PLEASE REMOVE IT.

Eighth paragraph: Additionally, (ADD "THE") majority of the included studies only (REMOVE ONLY) did not further specify the pathological findings aside from a diagnostic line, and this further limits a deeper analysis of these results.

Conclusion: Metastases can occur in patients with a history of malignancy that had recently been vaccinated. I SUGGEST ADDING THAT THIS IS RELATED TO THE UNDERLYING MALIGNANCY AND NOT TO THE VACCINE

Twelfth paragraph:  "In the two cases of florid lymphoid hyperplasia, Larkin et al. also reported that there was progressive transformation of germinal centers (ADD A COMMA HERE) while Tintle et al.34 also reported Langerhans cell hyperplasia."

Thirteenth paragraph: "There was no statistically significant difference in the duration from last vaccination to CL between those with reactive changes or negative for malignancy and those with florid lymphoid hyperplasia. The largest dimension of lymph node did not differ significantly amongst these three diagnoses." YOU SHOULD ADD NUMBERS HERE.

Fourteenth paragraph: All patients had a prior history of malignancy, with a history of melanoma in 42.1%, breast cancer in 36.8%, merkel cell carcinoma in 10.5%, and the other malignancies include (INCLUDED) cecum-appendix NET and renal cell carcinoma.

Sixteenth paragraph: Abnormal PET/CT findings include FDG uptake or hypermetabolic lymph nodes while abnormal US findings include thickened cortex. Abnormal CT/MRI findings include a length/width ratio of less than 1.5, cortical thickening, and asymmetricity. INCLUDED IN BOTH SENTENCES!

PLEASE REVIEW THE ENTIRE PAPER - THERE ARE OTHER SITES WHERE YOU SHOULD CORRECT - INCLUDED AND NOT INCLUDE

Seventeenth paragraph: surgical resection could be excision biopsy or lymphadenectomy, please specify.

Eighteenth paragraph: Placke et al. also reported two cases of metastases in patients with a history of melanoma or merkel cell carcinoma which ultrasound of the SLDI could not exclude malignancy. REMOVE THE WORD ALSO, AND IMPROVE THIS SENTENCE - ONE PATIENT WITH A HISTORY OF MELANOMA AND ONE PATIENT WITH HISTORY OF MERCKEL CELL CARCINOMA (NOT "OR"), IN WHICH (NOT WHICH) THE ULTRASOUND FINDINGS COULD NOT EXCLUDE MALIGNANCY.

Nineteenth paragraph: Findings are summarised in Table 4. A total of 14 studies that reported pathological findings in non-COVID-19 vaccine-associated lymphadenopathy was identified, (WERE INDENTIFIED)

Author Response

Reviewer 2

Thank you very much for inviting me to review this interesting paper. This are my suggestions:

ABSTRACT: line 2: widespread use in controlling the outbreak - add "of the vaccine" after the word use.

Add absolute number in the results section of the abstract. Please add a conclusion, you are only repeating the results section.

Authors’ response:

We thank the reviewer for the detailed comments.

The phrase, ‘of the vaccine’, has been added in page 1, line 25. The absolute numbers have been added in the results section of the abstract, in page 1, lines 37 to 39. We have added a more definitive conclusion, from page 1, line 49 to page 2 line 2, in addition to the original conclusion in page 1, lines 45 to 49.

INTRODUCTION: line 7, the comma after (mRNA) in not in place, should after the word vaccine.

Paragraph 3: You wrote:   Although some of these radiological findings may favour a reactive lymphadenopathy, studies have also reported suspicious ultrasound findings in lymphadenopathy after COVID-19 vaccination in patients being followed up (ADD HERE "FOR") skin cancer, raising a diagnostic conundrum between a reactive and malignant process (A PERIOD IS MISSING) - YOU SHOULD MENTION ALSO BREAST CANCER, FOLLOW UP BUT MOSTLY SCREENING FOR BREAST CANCER, WHICH INVOLVES A GREAT NUMBER OF HEATHY WOMEN - ADD REFERENCES FOR THIS.

Last paragraph of introduction: Histopathological and cytopathological findings, obtained through procedures including fine needle aspiration and biopsy (ADD "CORE NEEDLE" BEFORE THE WORD BIOPSY)

Authors’ response:

The comma has been rectified in page 2, line 11.

The ‘for’ comment has been rectified in page 3, line 7.

A full stop has been added in page 3, line 8.

The breast cancer comment has been addressed in page 3, lines 8 to 10.

The ‘core needle’ comment has been addressed in page 3, line 13.

MATERIALS AND METHODS:

First paragraph: 

"A literature (I THINK THE WORD "SEARCH" IS MISSING HERE) was performed, in accordance with the PRISMA statement,19 through PubMed and Google Scholar, on 11 December 2021."

"The reference lists of full-text articles were search for relevant studies that were not identified in the initial search." - PLEASE IMPROVE THIS SENTENCE, IT IS HARD TO UNDERSTAND.

Third paragraph: We extracted the pertinent details from each study, and these include: I THINK IT SHOULD BE "INCLUDED"

Last paragraph of materials and methods: "SLDI patients are (ADD THE WORD "DETECTED") on routine imaging follow-up for conditions including a history of malignancy." AGAIN, ADD HERE SCREENING (SINCE SCREENING FOR BREAST CANCER ACCOUNTS FOR A GREAT NUMBER OF THESE PATIENTS - ULTRASOUND PERFORMED AS SCREENING FOR BREAST CANCER INCLUDES THE AXILLAE.) JUST MENTION "CONDITIONS INCLUDING A HISTORY OF MALIGNANCY, OR SCREENING FOR CANCER' SUCH AS BREAST CANCER.

Authors’ response:

The ‘search’ comment is addressed in page 3, line 21.

The ‘reference list’ comment is addressed in page 3, lines 25.

The ‘included’ comment is addressed in page 3, line 39.

The ‘detected’ comment is addressed in page 4, line 6.

The last comment is addressed in page 4, lines 8 to 10.

RESULTS:

Second paragraph: Most studies are case reports (n = 12, 48.0%), followed by case series (n = 9, 36.0%) (A COMMA IS MISSING HERE) retrospective studies (n = 3, 12.0%), and not reported in one study (4.0%).

Authors’ response: A comma has been added in page 4, line 36.

Third paragraph: Pooled analysis of 21 studies included 37 patients, with a mean age of 47.8 ± 19.1 years (ADD THE WORD "OLD" HERE) and 23 (62.2%) (ADD THE WORD "PATIENTS") were females.

Authors’ response: The comment has been addressed in page 4, line 46.

ADD ABSOLUTE NUMBER AND NOT ONLY PERCENTAGES WHERE IT IS MISSING.

Authors’ response: This has been addressed throughout the results section.

Fifth paragraph: Last sentence - The other procedures that were performed include surgical resection and complete lymphadenectomy - I THINK IT SHOULD BE "INCLUDED"

Authors’ response: This has been addressed in page 5, line 31.

Sixth paragraph: Most cases of lymphadenopathy that were sent for pathological examination, were diagnosed as reactive or negative for malignancy (75.5%) (ADD A COMMA HERE) while florid lymphoid hyperplasia was reported in 5.4%.

Authors’ response: The sentence has been reconstructed for to improve its readability, in page 5, lines 36 to 39.

Seventh paragraph: A total of 15 studies reported pathological findings of COVD-19 vaccine-associated CL. (YOU HAVE A TYPO IN COVID-19

Authors’ response: This has been addressed in page 5, line 43.

YOU INCLUDED STUDY REFERENCE NUMBER 22 AS CLINICAL LYMPHADENOPATHY, BUT THIS IS SUBCLINICAL LYMPHADENOPATHY (IT IS CLEARLY WRITTEN THAT PATIENTS THAT HAD A BIOSPY HAD THEIR LYMPHADENOPATHY DETECTED ON IMAGING, WHICH MEANS THIS IS SUBCLINICAL). THIS STUDY SHOULD BE IN THE GROUP ANALYSIS OF SUBCLINICAL LYMPHADENOPATHY - PLEASE CORRECT THIS, AND REVIEW THE OTHER STUDIES THAT YOU INCLUDED IN EACH SUBGROUP TO ENSURE THEY ARE IN THE RIGHT SUBGROUP. PLEASE ADD IN THE TABLES IN EACH STUDY WHAT WAS THE CLINICAL FINDING THAT LED YOU TO CLASSIFY IT AS CLINICAL LYMPHADENOPATHY (CLINICAL SHOULD BE PALPABLE LYMPH NODES OR PAINFUL LYMPH NODES). IF LYMPHADENOPATHY WAS DETECTED ON IMAGING, THIS SHOULD BE CONSIDERED SUBCLINICAL. REVIEW ALL STUDIES FOR THAT PLEASE. IN ADDITION, YOU HAVE IN THAT STUDY US FINDINGS AND LARGEST DIMENSION OF LN (YOU WROTE AS NR), SO REVIEW AS WELL AND ALSO IN OTHER STUDIES.

Authors’ response: Reference number 22 included patients with both CL and SLDI, of which only 7 patients had a pathological diagnosis. The authors did not specify whether these 7 patients had CL or SLDI, and the US findings were not specified for these 7 patients. Hence, the US findings and largest dimension of LN were stated as ‘NR’. This study was not included in the group analysis. It was only included in the summary of the studies. We have clarified that Faermann et al. included patients with CL and SLDI in page 5, lines 51 to 52. The clinical finding that led to the classification of CL has also been added in Table 2, except the last study (Tan et al.) that did not specify whether it was palpable and/or painful lymphadenopathy, although it was detected clinically prior to imaging.

DISCUSSION:

Third paragraph: For (CHANGE FOR "AMONG PATIENTS") patients with SLDI, although most cases were diagnosed as reactive or negative for malignancy, granulomatous inflammation and metastases were also reported in a proportion of this population. HOW MANY PATIENTS HAD METASTASIS? ADD HERE HOW MANY AND IF ONLY ONE, ADD THAT IT WAS RARE AND ALTHOUGH RARE, YTHIS DD SHOULD BE CONSIDERED AS YOU MENTIONED.

Authors’ response:

The ‘among’ comment has been addressed in page 9, line 42.

The second comment has been addressed in page 9, lines 44 to 45.

Third paragraph: . Despite these associations of COVID-19 vaccine with lymphadenopathy that were diagnosed as KFD, granulomatous inflammation and metastases, it remains unclear whether these conditions are related or unrelated to the vaccine. The etiology and pathogenesis of KFD remains unclear, with viruses being postulated to be a key inciting agent. As such, based on findings from this study, we can only conclude the association of COVID-19 vaccine with these lymphadenopathies. PLEASE DON'T SAY THAT VACCINATION IS ASSOCIATED WITH LYMPH NODE METASTASIS!! YOU CANNOT CONCLUDE THIS AND IT IS NOT ASSOCIATED WITH THE VACCINE, BUT WITH THE UBDERLYING CANCER! REMOVE THIS ASSOCIATION AND REMOVE THIS LAST SENTENCE.

Authors’ response: This has been addressed in page 9, line 52 to page 10 line 2.

Sixth paragraph: Most of the cases WERE

Authors’ response: This has been addressed in page 10, line 43.

Seventh paragraph: second sentence is too long, please split it, and expain that effectiveness of vaccine was to REDUCE these outcomes.

Authors’ response: This has been addressed in page 11, lines 11 to 15.

With the likely need for future booster doses, lymphadenopathy after the third COVID19 vaccine has also already been reported6 and is expected to increase with time. - WHY ALSO? PLEASE REMOVE IT.

Authors’ response: This has been addressed in page 11, line 12 to 15.

Eighth paragraph: Additionally, (ADD "THE") majority of the included studies only (REMOVE ONLY) did not further specify the pathological findings aside from a diagnostic line, and this further limits a deeper analysis of these results.

Authors’ response: This has been addressed in page 11, lines 23 to 25.

Conclusion: Metastases can occur in patients with a history of malignancy that had recently been vaccinated. I SUGGEST ADDING THAT THIS IS RELATED TO THE UNDERLYING MALIGNANCY AND NOT TO THE VACCINE

Authors’ response: This has been addressed in page 11, lines 49 to 50.

Twelfth paragraph:  "In the two cases of florid lymphoid hyperplasia, Larkin et al. also reported that there was progressive transformation of germinal centers (ADD A COMMA HERE) while Tintle et al.34 also reported Langerhans cell hyperplasia."

Authors’ response: This has been addressed in page 6, line 49.

Thirteenth paragraph: "There was no statistically significant difference in the duration from last vaccination to CL between those with reactive changes or negative for malignancy and those with florid lymphoid hyperplasia. The largest dimension of lymph node did not differ significantly amongst these three diagnoses." YOU SHOULD ADD NUMBERS HERE.

Authors’ response: This has been addressed in page 7, lines 11 to 12.

Fourteenth paragraph: All patients had a prior history of malignancy, with a history of melanoma in 42.1%, breast cancer in 36.8%, merkel cell carcinoma in 10.5%, and the other malignancies include (INCLUDED) cecum-appendix NET and renal cell carcinoma.

Authors’ response: This has been addressed in page 7, line 23 to 26.

Sixteenth paragraph: Abnormal PET/CT findings include FDG uptake or hypermetabolic lymph nodes while abnormal US findings include thickened cortex. Abnormal CT/MRI findings include a length/width ratio of less than 1.5, cortical thickening, and asymmetricity. INCLUDED IN BOTH SENTENCES!

Authors’ response: This has been addressed in page 7, lines 47 to 49.

PLEASE REVIEW THE ENTIRE PAPER - THERE ARE OTHER SITES WHERE YOU SHOULD CORRECT - INCLUDED AND NOT INCLUDE

Authors’ response: This has been addressed throughout the paper.

Seventeenth paragraph: surgical resection could be excision biopsy or lymphadenectomy, please specify.

Authors’ response: The authors of that paper that mentioned surgical resection did not specify further.

Eighteenth paragraph: Placke et al. also reported two cases of metastases in patients with a history of melanoma or merkel cell carcinoma which ultrasound of the SLDI could not exclude malignancy. REMOVE THE WORD ALSO, AND IMPROVE THIS SENTENCE - ONE PATIENT WITH A HISTORY OF MELANOMA AND ONE PATIENT WITH HISTORY OF MERCKEL CELL CARCINOMA (NOT "OR"), IN WHICH (NOT WHICH) THE ULTRASOUND FINDINGS COULD NOT EXCLUDE MALIGNANCY.

Authors’ response: This has been addressed in page 8, lines 11 to 15.

Nineteenth paragraph: Findings are summarised in Table 4. A total of 14 studies that reported pathological findings in non-COVID-19 vaccine-associated lymphadenopathy was identified, (WERE INDENTIFIED)

Authors’ response: This has been addressed in page 8, line 18.

Reviewer 3 Report (New Reviewer)

Dear Authors,

First, I want to thank you for the paper you submitted. Finding an enlarged or 18FDG-avid lymph node in an oncological patient – diagnosed or suspected – always leads clinicians to a more thorough study to exclude a metastasis, the confirmation of which may change therapeutic approach. In daily clinical routine it is important to know the prevalence of “benign” lymphadenopathy due to other causes (such as vaccines) to avoid useless and/or risky diagnostic exams, evaluating correctly risk/benefit ratio. The study of yours, although the number of patients examined is little, is an interesting overview of principal characteristics of lymphadenopathy associated with COVID-19 vaccine. I don’t have to add anything to your brilliant analyses and conclusions, but I suggest you to check also these papers I will enlist you below to eventual further hints:

-          Bshesh, K., Khan, W., Vattoth, A. L., Janjua, E., Nauman, A., Almasri, M., Mohamed Ali, A., Ramadorai, V., Mushannen, B., AlSubaie, M., Mohammed, I., Hammoud, M., Paul, P., Alkaabi, H., Haji, A., Laws, S., & Zakaria, D. (2022). Lymphadenopathy post-COVID-19 vaccination with increased FDG uptake may be falsely attributed to oncological disorders: A systematic review. Journal of medical virology, 94(5), 1833–1845. https://doi.org/10.1002/jmv.27599;

-          Samkowski, J., Sklinda, K., & Walecki, J. M. (2022). Lymphadenopathy in the era of COVID-19 vaccination - an oncological dilemma in diagnostic imaging. Polish journal of radiology, 87, e304–e310. https://doi.org/10.5114/pjr.2022.117560

-          McIntosh, L. J., Bankier, A. A., Vijayaraghavan, G. R., Licho, R., & Rosen, M. P. (2021). COVID-19 Vaccination-Related Uptake on FDG PET/CT: An Emerging Dilemma and Suggestions for Management. AJR. American journal of roentgenology, 217(4), 975–983. https://doi.org/10.2214/AJR.21.25728

-          van Nijnatten, T., Jochelson, M. S., & Lobbes, M. (2022). Axillary lymph node characteristics in breast cancer patients versus post-COVID-19 vaccination: Overview of current evidence per imaging modality. European journal of radiology, 152, 110334. https://doi.org/10.1016/j.ejrad.2022.110334

All those papers were published after the period of your article collection.

As a side note (but it lies outside the aim of your paper), it would be interesting to collect systematically differences between images related to malignant lymph nodes versus COVID-19 vaccine-associated lymphadenopathy: maybe it could be a hint for further studies.

Author Response

Reviewer 3

Dear Authors,

First, I want to thank you for the paper you submitted. Finding an enlarged or 18FDG-avid lymph node in an oncological patient – diagnosed or suspected – always leads clinicians to a more thorough study to exclude a metastasis, the confirmation of which may change therapeutic approach. In daily clinical routine it is important to know the prevalence of “benign” lymphadenopathy due to other causes (such as vaccines) to avoid useless and/or risky diagnostic exams, evaluating correctly risk/benefit ratio. The study of yours, although the number of patients examined is little, is an interesting overview of principal characteristics of lymphadenopathy associated with COVID-19 vaccine. I don’t have to add anything to your brilliant analyses and conclusions, but I suggest you to check also these papers I will enlist you below to eventual further hints:

-          Bshesh, K., Khan, W., Vattoth, A. L., Janjua, E., Nauman, A., Almasri, M., Mohamed Ali, A., Ramadorai, V., Mushannen, B., AlSubaie, M., Mohammed, I., Hammoud, M., Paul, P., Alkaabi, H., Haji, A., Laws, S., & Zakaria, D. (2022). Lymphadenopathy post-COVID-19 vaccination with increased FDG uptake may be falsely attributed to oncological disorders: A systematic review. Journal of medical virology, 94(5), 1833–1845. https://doi.org/10.1002/jmv.27599;

-          Samkowski, J., Sklinda, K., & Walecki, J. M. (2022). Lymphadenopathy in the era of COVID-19 vaccination - an oncological dilemma in diagnostic imaging. Polish journal of radiology, 87, e304–e310. https://doi.org/10.5114/pjr.2022.117560

-          McIntosh, L. J., Bankier, A. A., Vijayaraghavan, G. R., Licho, R., & Rosen, M. P. (2021). COVID-19 Vaccination-Related Uptake on FDG PET/CT: An Emerging Dilemma and Suggestions for Management. AJR. American journal of roentgenology, 217(4), 975–983. https://doi.org/10.2214/AJR.21.25728

-          van Nijnatten, T., Jochelson, M. S., & Lobbes, M. (2022). Axillary lymph node characteristics in breast cancer patients versus post-COVID-19 vaccination: Overview of current evidence per imaging modality. European journal of radiology, 152, 110334. https://doi.org/10.1016/j.ejrad.2022.110334

All those papers were published after the period of your article collection.

As a side note (but it lies outside the aim of your paper), it would be interesting to collect systematically differences between images related to malignant lymph nodes versus COVID-19 vaccine-associated lymphadenopathy: maybe it could be a hint for further studies.

Authors’ response:

We thank the reviewer for the detailed comments. We have added in the suggested references to enhance our discussion, in page 10, lines 16 to 19 and lines 22 to 27.

Round 2

Reviewer 2 Report (New Reviewer)

Thank you very much for making the effort in addressing all the changes I've asked. Congratulations for the excellent work. I think the paper is now much better written and much better to read and understand, and important changes have been made. It is a very interesting paper and with a good value.

I have just a very few comments:

Results: 3.5: Subclinical lymphadenopathy detected on imaging (SLDI):

Sixth paragraph: "However, granulomatous inflammation and metastases 28 were reported in two cases respectively (2/19, 10.5%)." I AM NOTE SURE I UNDERSTAND WHY YOU WROTE RESPECTIVELY HERE, PLEASE REVIEW.

Discussion:

8th paragraph, line 10: I think it should be "further limits"

Author Response

Reviewer’s comments:

Thank you very much for making the effort in addressing all the changes I've asked. Congratulations for the excellent work. I think the paper is now much better written and much better to read and understand, and important changes have been made. It is a very interesting paper and with a good value.

I have just a very few comments:

Results: 3.5: Subclinical lymphadenopathy detected on imaging (SLDI):

Sixth paragraph: "However, granulomatous inflammation and metastases 28 were reported in two cases respectively (2/19, 10.5%)." I AM NOTE SURE I UNDERSTAND WHY YOU WROTE RESPECTIVELY HERE, PLEASE REVIEW.

Discussion:

8th paragraph, line 10: I think it should be "further limits"

Authors’ response:

We thank the reviewer for the comments.

For the comment regarding section 3.5, this has been addressed in page 8, lines 5 to 7.

For the comment regarding ‘further limits’, this has been addressed in page 11, line 25.

This manuscript is a resubmission of an earlier submission. The following is a list of the peer review reports and author responses from that submission.

Round 1

Reviewer 1 Report

This article aims to review and characterize the histo- and cytopathological findings in COVID-19 vaccine-associated lymphadenopathy and compare them with that of non-COVID-19 vaccine-related lymphadenopathy. The authors have performed the literature search and pooled analysis of many clinical variables. While a lot of details have been put into the writing of this manuscript, there are several important limitations to the study in its current form:

1. As cited by the authors as a limitation of their study, there is a paucity of specific histological or cytological findings that can be extracted from the majority of the included studies. As such, the analyses performed in this study focus mostly on clinical data available, and very little detail and comparison were made with reference to specific histological and/or cytological findings, which does not fit well with the proposed title of the study. 

2. It would be more apt to limit the comparison of histological and/or cytological findings of COVID-19 vaccine-associated lymphadenopathy with other viral vaccine-related causes. Half of the included studies on non-COVID-19 vaccine-associated lymphadenopathy in this manuscript are related to the BCG vaccine, which is known to evoke a very different immunological response (ie granulomatous inflammation). 

Minor comments:

1. The presentation of data feels repetitive and does not convey the key summaries succinctly. The information included in the summary tables can be improved.

2. Figure 1 can be improved. It is not clear why n=433 records were excluded. It is not clear what "Report not retrieved (n=0)" means. It is not clear what it means "No pathological findings n=67".

3. It appears that the authors missed out on some reports on non-COVID-19 viral vaccine-associated lymphadenopathy. For example, the following report has not been cited.

Podugu, A.; Kobe, M. Kikuchi-Fujimoto Disease (KFD): A Rare Cause of Fever and Lymphadenopathy Following Influenza Vaccination. Chest 2013, 144, 230A. 

4. The source of the images in Figure 2 has not been cited. If these images are merely used to illustrate the cytological features of reactive lymphoid hyperplasia, and are not obtained from a patient who has received the COVID-19 vaccine, this figure is felt to be unnecessary.

Reviewer 2 Report

The Authors carried a systematic review of histopathological and cytopathological findings in COVID-19 and non-COVID-19 vaccine-associated lymphadenopathies. The manuscript is well-written and detailed. This referee has a few suggestions:

It would be desirable to cut figure 2 for two reasons: a) it is not really representative (histopathology would be much more informative), b) a lymph node aspirate is not recommended by the WHO in case a lymphoproliferative disorder is suspected.

How Dotlic et al. excluded a peripheral T-cell lymphoma in their case? The search for TdT, CD34 and CD117 does not suffice for ruling out this possibility, suggested by cellular atypia and an exceedingly high proliferation rate. Did they perform molecular studies? If not, it would be wise to exclude the case.